# HiddenTables & PyQTax: A Cooperative Game and Dataset For TableQA to Ensure Scale and Data Privacy Across a Myriad of Taxonomies

**William Watson** and **Nicole Cho** and **Tucker Balch** and **Manuela Veloso**

J.P. Morgan AI Research

New York, NY, USA

{william.watson, tucker.balch, manuela.veloso}@jpmchase.com

nicole.cho@jpmorgan.com

## Abstract

A myriad of different Large Language Models (LLMs) face a common challenge in contextually analyzing table question-answering tasks. These challenges are engendered from (1) finite context windows for large tables, (2) multi-faceted discrepancies amongst tokenization patterns against cell boundaries, and (3) various limitations stemming from data confidentiality in the process of using external models such as `gpt-3.5-turbo`. We propose a cooperative game dubbed "HiddenTables" as a potential resolution to this challenge. In essence, "HiddenTables" is played between the code-generating LLM "Solver" and the "Oracle" which evaluates the ability of the LLM agents to solve Table QA tasks. This game is based on natural language schemas and importantly, ensures the security of the underlying data. We provide evidential experiments on a diverse set of tables that demonstrate an LLM's collective inability to generalize and perform on complex queries, handle compositional dependencies, and align natural language to programmatic commands when concrete table schemas are provided. Unlike encoder-based models, we have pushed the boundaries of "HiddenTables" to not be limited by the number of rows - therefore we exhibit improved efficiency in prompt and completion tokens. Our infrastructure has spawned a new dataset "PyQTax" that spans across 116,671 question-table-answer triplets and provides additional fine-grained breakdowns & labels for varying question taxonomies. Therefore, in tandem with our academic contributions regarding LLMs' deficiency in TableQA tasks, "HiddenTables" is a tactile manifestation of how LLMs can interact with massive datasets while ensuring data security and minimizing generation costs.

## 1 Introduction

Encoder-based approaches in contextually analyzing table question-answering tasks for language models typically prioritize and highlight the methods' achievement in accuracy (Herzig et al., 2020; Liu et al., 2022). However, in many cases, a prerequisite for these approaches to achieve such accuracy is the exposition of tabular content in its entirety and the indulgent ingestion of tokens (Herzig et al., 2020; Yin et al., 2020; Yu et al., 2021; Liu et al., 2022). Such liberal dispositions towards privacy and efficiency can be deemed as impractical in the tangible deployment process of language models within institutions. Moreover, such necessities to expose the underlying data begs the question of whether the model actually understands the question to provide an accurate answer. In essence, our endeavor is also an intellectual pursuit to answer the "chinese room argument" with regards to language models (Cole, 2023). Therefore, we propose an alternative approach for table question-answering tasks - a cooperative game dubbed "HiddenTables". HiddenTables is comprised of two agents: an "Oracle" and a "Solver", in which the latter generates code to answer user queries relying solely on the Oracle's instructions and relaying of schema. In other words, the game is played without the Solver knowing the tabular content. The Solver's code is then evaluated by the secure Oracle that relays the answer to the user or asks follow-up questions to the Solver. Figure 1 summarizes the environmental set-up that our method enables between a user and `gpt-3.5-turbo`.

Therefore this paper sets forth a general system architecture that can be employed across a myriad of taxonomies and tabular formats. We find that the accuracy of `gpt-3.5-turbo` has decreased with our cooperative game albeit with lesser tokens and tightened privacy. In summary, HiddenTables and its pertinent experiments have brought forth the following contributions to the academic community:

- We have devised a construct that can complement an encoder-based approach in table question-answering tasks for language mod-

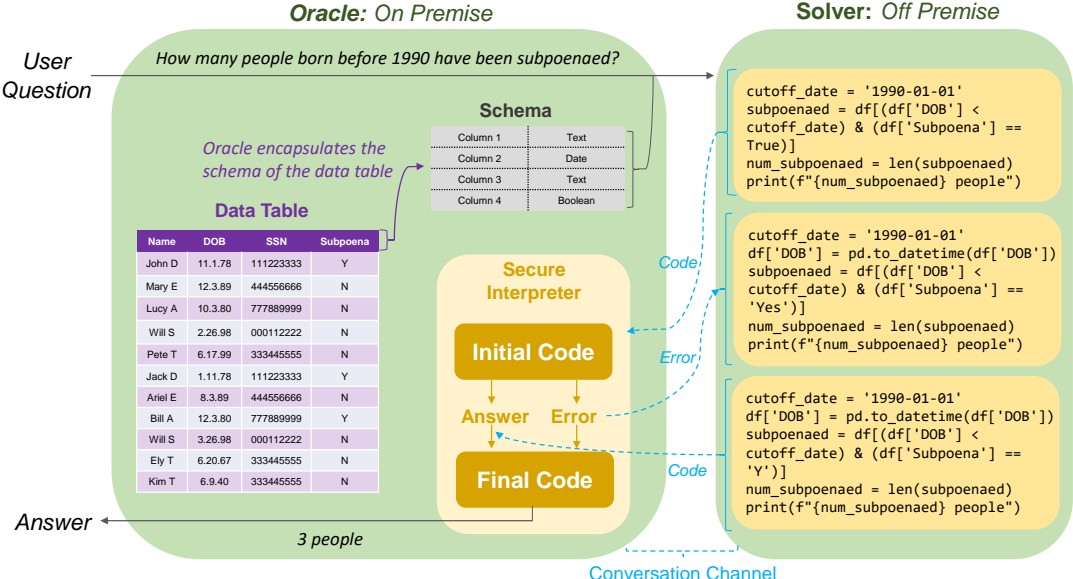

Figure 1: Overview of our system apparatus to encourage HiddenTables. The setup requires two agents, an **Oracle** and the **Solver**, which may or may not be on the same device. For our purposes, the Solver is a `gpt-3.5-turbo` LLM agent that handles generation off-site, and therefore potentially offers risk of adversarial attacks. We outline the **conversation** between our agents, which is a message-passing channel that transfers solution code along with follow-up questions, *without* exposing any information from the datalake. Finally, the Oracle will provide the answer to the user.

els, as a less costly and more secure alternative with significantly decreased risk in data exploitation.

- Leveraging code-generation capabilities of language models allows for a full chain of thought exposition via programmatic commands, enabling further interpretability into the answer retrieval process than what prior encoder or sequence-to-sequence models provided.

- Our cooperative game is a robust demonstration that the accuracy of `gpt-3.5-turbo` decreases rapidly when language models are not given the entirety of the data yet improves with consecutive rounds of feedback.

- Therefore, our study contributes to not only the institutional adoption process of language models but also the critical question of general intelligence capabilities of language models with regards to table question-answering tasks.

- Additionally, HiddenTables has generated a new dataset "PyQTax" that encompasses 116,671 question-table-answer-python quadruplets of varying degrees and taxonomies for promising future academic experiments.

## 2 Related Work

Since the advent of Transformer-based attention models, pre-trained language models have shown remarkable success in learning and encoding the semantics of tabular content (Vaswani et al., 2017). Methods employing encoder-based architectures rely on Masked Language Modeling (MLM) to learn semantics and dense representations of tabular content. Yet they are pre-trained on natural language text tokenized by byte-pair-encoding or WordPiece (Devlin et al., 2019; Sennrich et al., 2016) which can misalign with tabular structure. TaPaS (Herzig et al., 2020) employed an encoder that is pre-trained with whole word masking, TaBERT (Yin et al., 2020) leveraged Masked Column Prediction and Cell Value Recovery to learn structure, and GraPPa (Yu et al., 2021) augmented pre-training with synthetic SQL to inject structural properties into the model. In contrast to these encoder-based approaches, TaPEx (Liu et al., 2022) relies on a BART encoder-decoder backbone (Lewis et al., 2019) to encode tables and generate answers in an autoregressive fashion.

However, HiddenTables relies solely on the generative power of autoregressive decoders (Brown et al., 2020) and instruction-aligned models trained with reinforcement learning from human feedback

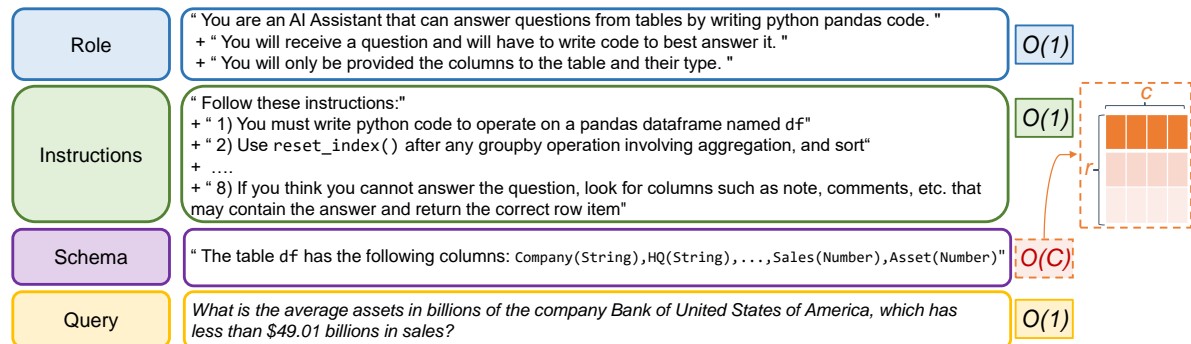

Figure 2: Outline of our **Role**, **Instructions**, **Schema**, and **Question** (**RISQ**) prompt template that the Oracle generates for the Solver. Each instruction was curated to align the Solver's code to work with our tables. For instance, all string comparisons are case insensitive and Unicode normalized. For each prompt component we outline the token complexity, which is bounded by the number of columns $O(c)$ in the schema.

(RLHF) (Ouyang et al., 2022), to generate solutions based on prompts (Liu et al., 2023) rather than fine-tuning.

Furthermore, prior work shows that language models can more effectively solve problems when decomposing them into steps or a chain of thought (Wei et al., 2023; Nye et al., 2021). HiddenTables is inspired by using chain of thought through code, as demonstrated by (Liang et al., 2022) for robotic programs, action plan generation in robotics (Ahn et al., 2022), web browsing (Nakano et al., 2022), tool APIs (Schick et al., 2023), or the generation of valid programs for arithmetic computation (Gao et al., 2022). ReAct (Yao et al., 2023) explores how LLMs can improve their chain of thought reasoning via intermediate actions and interactions with external sources. Furthermore, BINDER (Cheng et al., 2023) demonstrated a neural-symbolic approach to mapping questions to a program, building upon the work in (Rajkumar et al., 2022) for semantic parsing and code generation.

Also, previous literature has explored how LLMs can interact with themselves through intermediate followups (Press et al., 2023), chained LLM prompts (Wu et al., 2022), or cascades (Dohan et al., 2022). (Reynolds and McDonell, 2021) has proposed how LLMs can be encouraged to generate their own prompts for solving tasks. Finally, MemPrompt (Madaan et al., 2022) demonstrated that memories of errors and user feedback can be incorporated as part of the conversation to help prevent repetitive mistakes.

## 3 Methodology

Our proposed framework is inspired from the "chinese room argument" - *to what extent could lan-* *guage models truly comprehend natural language and align language to the correct solution when only given the table schema?* In HiddenTables, two agents exist: the **Oracle** and the **Solver**. The clear delineation between these two agents' respective roles not only allows the user to test the model's holistic ability to comprehend tabular content but also enables the preservation of privacy with regards to the underlying data on-premise. In this context, our proposed apparatus allows the two agents to engage in a *conversation*, in which the Oracle may ask questions and the Solver will generate code that could solve the Oracle's question. Next, the Oracle will evaluate and follow-up which enables the Solver to correct any mistakes or misunderstandings. This game is played for a maximum of seven rounds to prevent infinite cycles between the agents. Throughout this process, no data entries are exposed to the Solver - the Solver must produce executable code relying solely on the schema and the set of instructions.

### 3.1 The Oracle

The Oracle takes the user query and crafts an appropriate prompt for the Solver, which is structured as a **role**, **instruction**, relevant **schema**, and the **question** (**RISQ**)[1]. It *will not* expose any individual data entries in the table. This allows the Oracle to protect highly confidential information in a firewalled system from any adversaries. This prompt is then sent to the Solver, which is fully outlined in Figure 2. Furthermore, we include a discussion on the prompt burden (§3.8) juxtaposed against holistic encoder methods (Table 1).

---

[1]Further details about our RISQ prompt can be found in Appendix §C.

| Dataset | Split | Samples | Query | Table | Answer | Total |
|---|---|---|---|---|---|---|
| WikiSQL | Train | 56,355 | 913,649 | 31,357,980 | 229,264 | 32,500,893 |
| | Val | 8,421 | 137,067 | 4,399,557 | 33,147 | 4,569,771 |
| | Test | 15,878 | 258,362 | 9,562,411 | 65,560 | 9,886,333 |
| | Total | 80,654 | 1,309,078 | 45,319,948 | 327,971 | 46,956,997 |
| WikiTable Questions | Train | 11,321 | 146,080 | 9,180,294 | 57,677 | 9,384,051 |
| | Val | 2,831 | 36,525 | 2,153,166 | 14,448 | 2,204,139 |
| | Test | 4,344 | 55,385 | 3,343,142 | 21,538 | 3,420,065 |
| | Total | 18,496 | 237,990 | 14,676,602 | 93,663 | 15,008,255 |
| SQA | Train | 12,276 | 126,536 | 4,907,917 | 386,288 | 5,420,741 |
| | Val | 2,265 | 23,113 | 947,744 | 70,886 | 1,041,743 |
| | Test | 3,012 | 29,180 | 1,262,051 | 106,919 | 1,398,150 |
| | Total | 17,553 | 178,829 | 7,117,712 | 564,093 | 7,860,634 |
| **Grand Totals** | | **116,703** | **1,725,897** | **67,114,262** | **985,727** | **69,825,866** |

Table 1: Number of Tokens required to be analyzed by `gpt-3.5-turbo` if a holistic table encoding approach was adopted, as in (Herzig et al., 2020; Liu et al., 2022). Query, Table and Answer totals are provided per dataset and in aggregate. Note that the largest table dimensions encountered were 1,956 rows, 44 columns, and 11,600 entries. Our system seeks to minimize the token usage through schemas only - therefore bounding the number of tokens used to the number of columns, instead of to the number of entries.

The Oracle also maintains the datalake in the Secure Interpreter, that executes the code produced by the Solver (§3.2). Moreover, the Secure Interpreter ensures that any request to expose the dataset via code injections is rejected and that it only returns the answer to the user's query. We provide more details into the Oracle's followups in Section §3.3.

## 3.2 The Solver

The Solver is a code-generating LLM agent that accepts the Oracle's instructions, question, and tabular schema. Then, it strives to translate and align the prompt into a sequence of executable operators that can be applied to the hidden table. In prior literature, the main choice of query language was SQL (Zhong et al., 2017); however, within our construct, the Solver does not need to be restricted to any specific programming language. HiddenTables opted to use Python as the Solver's language of choice, as it is dynamically typed, easily readable, and procedure-oriented. Therefore, it is convenient to view the chain-of-thought through iterative commands. Finally, byproducts of our generative experiments have yielded an amalgamation of verified python programs grounded to each question-table-answer triplet that are linked to varying taxonomies - we introduce this new dataset as PyQTax (§3.9).

## 3.3 The Conversation

We now outline the communication channel between the two agents. Foremost, the Oracle sends the instructions to the Solver. The instructions are an itemized list that dictates the format of the Solver's response. The instructions and rationale are outlined in Figure 2.

Next, the Solver responds with what it deems to be the best sequence of commands to answer the query. This is sent to the Oracle as free-text along with embedded code, including artifacts pertaining to explanations and chain of thought. Consequently, the Oracle sets up a secure environment, locally fire-walled with its dataset. Aforementioned, this environment ensures that any arbitrary execution of code is non-destructive and any exposure of the underlying tabular data is disabled.

As a result of this conversation, there are two states that will be defined in detail - a state of "successful retrieval" or one of "failure". A state of successful retrieval is defined as one in which an answer has been generated from executing the Solver's code in the Oracle's secure environment. This answer could be a text entry from the table, an aggregated value such as sum, or a list of table entries. In contrast, a state of failure is defined as an error message, such as `Value` or `Index` errors, `NULL` answers that provide no identifiable answer (empty dataframes) nor any executable code, or the Solver's comment that it cannot answer. For each type of failure, the Oracle handles the state differently. Firstly, errors can be sanitized to remove any data references and fed back to the Solver, as prior literature regarding self-correcting code has

discussed (Madaan et al., 2022). Secondly, empty dataframes can be conveniently identified with the Solver being informed that the generated code produced no valid results. Thirdly, if the Solver is conservative in answering the question and provides no executable code, the Oracle reassures the Solver that the question can be answered from the table provided. Within this context, new failures can be re-prompted to the Solver for correction by the Oracle.

With this apparatus to correct initial failures while retaining the original context throughout the conversation session, we allow the Oracle and Solver to interact for a maximum of seven times before the conversation is halted and the final verdict for this query is designated as a failure. We have discovered that failures are common when the answer is within an extractive span in a single table cell (free text) or if the answer resides in a generic column such as 'comments' or 'notes' that complicates contextual inference.

### 3.4 Minor Roles

**The User**    The user's query initiates our game of HiddenTables.

**Datalake**    The Oracle has read-access to a data-lake, which stores the tables and entries in a secured environment *on-premise*.

**Firewall**    This is a boundary to denote in Figure 1 - the *on-premise* and *off-premise* environments the agents operate in. This setup can enable guided entry into the *on-premise* environment.

### 3.5 Benefits of Demarcating the Roles

Demarcating the boundaries between the Oracle and Solver is to ensure that the underlying dataset is protected. This can be beneficial because firstly, for many institutions that handle sensitive or confidential data such as personally identifiable information, the Oracle can prevent any off-premise entities from accessing the data but still help generate answers. Secondly, this demarcation ensures that code is executed in a regulated and structured manner, regardless of the user's location or device. Thirdly, an additional layer of control has been generated, while still allowing third-party API providers to operate on the data.

### 3.6 Question, Table, and Answer Token Counts

Table 1 outlines each set's total token count for `gpt-3.5-turbo` if sent to the model. The number of tokens were determined by OpenAI's fast BPE encoder tiktoken[2]. The dominating term for token counts is in the table entries themselves - **96.1%** of the outstanding burden is located here. However, previous encoder-based methods were limited by the model's sequence length and memory constraint in computing multi-headed attention between every cell (Liu et al., 2022). In contrast, our construct is comparably linear in its token usage. If we define the number of rows as $r$ and the number of columns as $c$, then the total token count for a table is polynomial $O(rc)$, which is quadratic in time complexity as either term increases. However, in HiddenTables, since the only dependent variable required for solving a table query is bounded by the number of columns $O(c)$, token growth is linear. Each table could add $c \times r$ many rows - yet our task will still include the same number of columns $c$.

### 3.7 Privacy

Another by-product of this setup is **privacy**. Since row entries are omitted and safe guarded by the Oracle, the Solver must form a general solution from the schema only. More importantly, the Oracle can be configured with additional safety prompts and code policies to ensure that any adversarial attacks by the Solver are properly handled. However, this system may potentially need additional safeguards against side-channel attacks to obfuscate successful retrievals from failures (Kocher, 1996).

### 3.8 Prompt Burden

Given the replacement of table entries with our **RISQ** system prompt, we analyzed the distribution of usage tokens for all three of our datasets. Of the $116,661$ samples accepted by the Solver and responded to (without error), the average prompt burden was 279 tokens with a standard deviation of 19 tokens. The minimum, median, and maximum prompt usage was 243, 275, and 630, respectively. Overall, the total amount of tokens used in the Solver's system prompt was $32,546,634$. This is only $48.5\%$ of the burden incurred by using the entire table. As mentioned in §3.6, our construct is efficient for large tables with many rows, as the token burden remains constant for each new row of

---

[2]https://github.com/openai/tiktoken

data. Our Solver generated an average 115 tokens per answer, with a standard deviation of 61 tokens.

### 3.9 PyQTax

HiddenTables has produced PyQTax that aligns 116,671 question-table-answer triplets to Python code. In addition, PyQTax categorizes every question into varying taxonomies, such as difficulty, table size, question type, operator, and sequence length (for SQA). With these two additions, further research can be conducted into bolstering low-performing taxonomies and improving LLM code generalization in HiddenTables with Python.

## 4 Datasets

**WikiSQL ([Zhong et al., 2017](#))** The original purpose of WikiSQL was to translate natural language into SQL, and we have repurposed this task to write Python code. WikiSQL is comprised of simple questions - selecting and filtering table entries ($71.8\%$) that align well with the table schema. Aggregation operations only comprise $28.2\%$ of the questions. It consists of $80,654$ total examples over $24,241$ tables. However, $2\%$ of the set's answers are incorrect according to ([Herzig et al., 2020](#); [Liu et al., 2022](#)).

**WikiTQ ([Pasupat and Liang, 2015](#))** WikiTableQuestions is a more complex question-answering set sampled from tables in Wikipedia. It consists of $18,496$ examples over $2,108$ tables. Annotators were tasked with composing a series of complex questions involving operations such as comparisons, superlatives, aggregation, and arithmetic to create a challenging QA dataset.

**SQA ([Iyyer et al., 2017](#))** Building upon WikiTQ, SQA decomposes compositional questions into sequential orderings, in which each resulting question can be answered by one or more table cells. The main distinguishing factor for SQA is that the questions are conversational, built up from prior queries. The set consists of $17,553$ examples, $982$ tables, and $6,066$ sequences with an average sequence length of $2.06$ questions. The median sequence length is $2$ and the maximum is $8$ questions.

## 5 Analysis & Discussion

### 5.1 Table Size

We breakdown our analysis based upon the interquartile range on table tokens - small tables represent the lower quartile ($\approx 25\%$), average tables

| Difficulty | Count | Train | Val | Test |
|---|---|---|---|---|
| **Easy** | 618 | 78.7 | 75.0 | 82.4 |
| **Medium** | 55,659 | 72.3 | 72.5 | 72.2 |
| **Hard** | 19,303 | 59.2 | 60.1 | 60.0 |
| **Extra Hard** | 5,044 | 55.0 | 55.3 | 55.5 |
| **Operator** | Count | Train | Val | Test |
| SELECT | 57,923 | 72.1 | 72.9 | 72.1 |
| COUNT | 7,347 | 61.5 | 58.5 | 60.4 |
| MAX | 4,639 | 56.7 | 55.4 | 54.5 |
| MIN | 4,631 | 57.8 | 60.3 | 59.2 |
| AVG | 3,135 | 43.2 | 44.4 | 44.2 |
| SUM | 2,949 | 68.0 | 67.9 | 71.7 |
| **Table Size** | Count | Train | Val | Test |
| **Small** | 20,332 | 68.7 | 70.2 | 68.1 |
| **Average** | 39,799 | 68.9 | 68.6 | 68.5 |
| **Large** | 20,493 | 66.2 | 66.7 | 67.3 |
| **# Rows** | Count | Train | Val | Test |
| **Small** | 22,985 | 68.5 | 69.8 | 68.9 |
| **Average** | 38,263 | 68.3 | 68.1 | 67.3 |
| **Large** | 19,376 | 67.5 | 67.8 | 68.9 |
| **# Columns** | Count | Train | Val | Test |
| **Small** | 31,192 | 70.6 | 72.1 | 70.3 |
| **Average** | 35,072 | 68.8 | 68.6 | 68.3 |
| **Large** | 14,360 | 61.5 | 60.3 | 62.7 |
| **Conv Rnds** | Count | Train | Val | Test |
| **Round 1** | 62,478 | 79.5 | 79.5 | 78.9 |
| **Round 2** | 10,247 | 37.0 | 36.0 | 36.9 |
| **Round 3** | 3,108 | 36.7 | 33.3 | 39.8 |
| **Round 4** | 785 | 28.3 | 27.1 | 23.2 |
| **Round 5** | 536 | 30.9 | 29.8 | 30.9 |
| **Round 6** | 243 | 20.7 | 11.5 | 27.9 |
| **Round 7** | 3,227 | 1.5 | 3.2 | 1.3 |
| **All** | 80,624 | 68.2 | 68.5 | 68.1 |

Table 2: We provide breakdowns of each **WikiSQL**, split by complexity of the required operations to produce the answer and by each aggregator. The best performing taxonomies are **Easy** and **Medium** difficulty questions, SELECT, and tables with a small amount of columns. **Medium** style questions comprise 69% of the overall set, with hard at 24%. SELECT is the dominant operator at 71.8% of questions. TaPEx achieved a denotation accuracy of 89.5% on WikiSQL-Weak.

the middle $50\%$, and large tables the upper quartile ($\approx 75\%$). This enables outlier categorization into the pertinent buckets that guide the amount of content any model processes to produce an answer. For WikiSQL, the first and third quartiles are 247 and 607 tokens. For WikiTQ, the quartiles are 288 and 805. For SQA, the quartiles are 248 and 492.

| Operator | Count | Train | Val | Test |
|---|---|---|---|---|
| **Aggregate** | 4,854 | 45.6 | 44.1 | 44.1 |
| **Filter** | 4,418 | 39.9 | 42.9 | 40.6 |
| **Superlative** | 3,048 | 37.0 | 34.5 | 41.9 |
| **Comparative** | 2,497 | 42.6 | 35.1 | 42.8 |
| **Select** | 1,943 | 36.6 | 45.8 | 37.1 |
| **Arithmetic** | 1,562 | 32.0 | 31.2 | 33.2 |
| **Other** | 107 | 46.0 | 54.2 | 45.0 |
| **Group** | 67 | 35.3 | 41.7 | 57.1 |
| **Table Size** | **Count** | **Train** | **Val** | **Test** |
| **Small** | 4,505 | 43.7 | 46.7 | 45.5 |
| **Average** | 9,111 | 40.1 | 39.7 | 38.9 |
| **Large** | 4,870 | 36.9 | 38.0 | 41.0 |
| **# Rows** | **Count** | **Train** | **Val** | **Test** |
| **Small** | 5,555 | 39.8 | 42.0 | 43.4 |
| **Average** | 8,355 | 41.4 | 39.2 | 40.5 |
| **Large** | 4,576 | 38.6 | 40.3 | 39.8 |
| **# Columns** | **Count** | **Train** | **Val** | **Test** |
| **Small** | 7,346 | 40.6 | 38.3 | 40.1 |
| **Average** | 7,439 | 40.7 | 42.4 | 42.0 |
| **Large** | 3,701 | 38.5 | 40.8 | 41.4 |
| **Conv Rnds** | **Count** | **Train** | **Val** | **Test** |
| **Round 1** | 11,031 | 53.7 | 53.9 | 55.4 |
| **Round 2** | 3,258 | 32.4 | 31.9 | 31.3 |
| **Round 3** | 1,129 | 25.4 | 27.2 | 22.7 |
| **Round 4** | 441 | 20.4 | 17.5 | 14.4 |
| **Round 5** | 192 | 20.5 | 29.6 | 23.7 |
| **Round 6** | 121 | 22.5 | 20.0 | 15.4 |
| **Round 7** | 2,314 | 1.3 | 0.5 | 1.8 |
| **All** | 18,496 | 40.2 | 40.3 | 41.1 |

Table 3: We provide breakdowns of each **WikiTQ**, split by the type of operation, table size by entries, rows, and columns, and the number of conversation rounds required by the Solver. WikiTQ provides insight into how language models can handle complex QA challenges. We employ few-shot categorization to label each question (§5.4). The best performing taxonomies are **Aggregate** and **Comparative** for operators, small tables with limited entries, and tables solved in **Round 1**. Note that the Solver is consistent in performance regarding row size. TaPEx acheived a denotation accuracy of 57.0% and 57.5% on the Dev and Test set respectively.

We follow the same procedure for the number of table rows and columns, relying on the interquartile range to delineate small, average, and large tables. For WikiSQL, our quartiles for rows are $Q_1 = 7$, $Q_3 = 18$. For WikiTQ, our quartiles for rows are $Q_1 = 10$, $Q_3 = 25$. For SQA, our quartiles for rows are $Q_1 = 9$, $Q_3 = 17$. The first and third

| View | Count | Train | Val | Test |
|---|---|---|---|---|
| $Q_1$ | 6,065 | 51.8 | 51.5 | 58.0 |
| $Q_2$ | 6,064 | 33.8 | 31.8 | 32.2 |
| $Q_3$ | 4,035 | 34.0 | 35.3 | 38.7 |
| $Q_4$ | 1,106 | 37.3 | 36.0 | 47.6 |
| $Q_5$ | 222 | 35.1 | 60.0 | 42.1 |
| $Q_6$ | 38 | 32.1 | 0.0 | 44.4 |
| $Q_7$ | 15 | 18.2 | - | 100.0 |
| $Q_8$ | 6 | 16.7 | - | - |
| **Operator** | **Count** | **Train** | **Val** | **Test** |
| **Aggregate** | 596 | 31.8 | 34.5 | 36.9 |
| **Filter** | 3,860 | 40.3 | 43.4 | 46.2 |
| **Superlative** | 3,887 | 42.3 | 37.4 | 45.5 |
| **Comparative** | 4,269 | 38.4 | 42.8 | 41.6 |
| **Select** | 1,685 | 36.6 | 34.3 | 45.2 |
| **Arithmetic** | 273 | 37.9 | 24.0 | 41.5 |
| **Other** | 87 | 52.2 | 50.0 | 29.6 |
| **Group** | 20 | 45.5 | - | 55.6 |
| **N/A** | 2,874 | 42.7 | - | - |
| **Table Size** | **Count** | **Train** | **Val** | **Test** |
| **Small** | 4,545 | 42.1 | 42.6 | 48.0 |
| **Average** | 8,315 | 40.5 | 40.3 | 41.0 |
| **Large** | 4,691 | 38.1 | 37.4 | 44.2 |
| **# Rows** | **Count** | **Train** | **Val** | **Test** |
| **Small** | 5,144 | 41.1 | 42.5 | 45.2 |
| **Average** | 8,202 | 39.5 | 38.2 | 41.3 |
| **Large** | 4,205 | 41.0 | 40.3 | 46.7 |
| **# Columns** | **Count** | **Train** | **Val** | **Test** |
| **Small** | 7,430 | 41.1 | 38.1 | 45.2 |
| **Average** | 7,259 | 39.2 | 42.8 | 42.8 |
| **Large** | 2,862 | 40.9 | 39.0 | 43.5 |
| **Conv Rnds** | **Count** | **Train** | **Val** | **Test** |
| **Round 1** | 10,152 | 50.6 | 49.6 | 53.6 |
| **Round 2** | 3,574 | 43.9 | 46.7 | 45.7 |
| **Round 3** | 1,019 | 25.6 | 30.2 | 27.7 |
| **Round 4** | 358 | 17.4 | 21.3 | 21.3 |
| **Round 5** | 161 | 17.9 | 18.2 | 9.1 |
| **Round 6** | 111 | 21.1 | 21.1 | 43.8 |
| **Round 7** | 2,176 | 0.7 | 0.3 | 0.9 |
| **All** | 17,551 | 40.3 | 40.0 | 43.9 |

Table 4: Experimental results for **SQA** for all sequence lengths, operators, table sizes, and conversation length. Accuracy is reported only when applicable. Categorizations are reused from WikiTQ and **N/A** otherwise. Solver performance is strong on $Q_1$, **Filter**, **Superlative**, **Other** operators, and conversation rounds **1 & 2**. The **Arithmetic** and **Select** operators are the most deficient as compositional errors propagate downstream. TaPEx achieved SQA test accuracy of 74.5%

| | Scenario | | Conversation Rounds | | | | | | |
| Dataset | Split | Count | R1 | R2 | R3 | R4 | R5 | R6 | R7 |
|---------|-------|-------|-----|-----|-----|-----|-----|-----|-----|
| **WikiSQL** | Train | 56,335 | 61.4 | 66.1 | 67.6 | 67.8 | 68.0 | 68.1 | 68.2 |
| | Val | 8,417 | 62.2 | 66.7 | 67.9 | 68.2 | 68.4 | 68.4 | 68.5 |
| | Test | 15,872 | 61.4 | 66.0 | 67.5 | 67.8 | 68.0 | 68.1 | 68.1 |
| | Total | 80,624 | 61.5 | 66.2 | 67.6 | 67.9 | 68.1 | 68.1 | 68.2 |
| **WikiTQ** | Train | 11,313 | 31.7 | 37.5 | 39.1 | 39.7 | 39.9 | 40.1 | 40.2 |
| | Val | 2,831 | 32.6 | 38.1 | 39.5 | 39.9 | 40.2 | 40.3 | 40.3 |
| | Test | 4,342 | 33.6 | 38.9 | 40.3 | 40.6 | 40.8 | 40.9 | 41.1 |
| | Total | 18,496 | 32.3 | 37.9 | 39.5 | 39.9 | 40.2 | 40.3 | 40.5 |
| **SQA** | Train | 12,274 | 28.0 | 38.0 | 39.6 | 39.9 | 40.1 | 40.2 | 40.3 |
| | Val | 2,265 | 27.7 | 37.4 | 39.2 | 39.6 | 39.8 | 40.0 | 40.0 |
| | Test | 3,012 | 33.5 | 41.7 | 43.1 | 43.4 | 43.5 | 43.8 | 43.9 |
| | Total | 17,551 | 29.5 | 38.6 | 40.1 | 40.5 | 40.6 | 40.8 | 40.9 |

Table 5: Ablation results for the cumulative accuracy gains per additional conversation round. Each round includes the cumulative total of correct solutions, even if the conversation ended prematurely. Incremental gains in accuracy level off after the third conversation round, as a consequence of a dwindling pool of remaining unsolved problems. Furthermore, issues from parsing persist in the later conversation rounds as the Solver struggles to find the right formats or forgets the original task.

column quartiles were $Q_1 = 5$, $Q_3 = 7$ for all three datasets. Our experiments show that demarcations for columns show the largest differentials in performance favoring small tables, while our Solver is consistent across any number of rows. It is difficult to generalize the performance regarding table entries since the size is obfuscated by either the number of rows or columns.

## 5.2 Conversation Length & Cumulative Accuracy

For all datasets, we show the necessary number of attempts to write fully executable code. Our experiments show that while the probability of a successful retrieval decreases with more rounds, a considerable number of samples are being solved *correctly* in each round. As reported in Table 5, HiddenTables sees significant cumulative increases in the Solver's accuracy when paired with a Oracle agent for the first three conversation rounds. Afterwards, additional rounds yield very diminished accretive benefits.

## 5.3 WikiSQL

**SQL Query Difficulty** Following a similar analysis by (Yu et al., 2018; Liu et al., 2022), we breakdown our WikiSQL results by difficulty, yielding insights into how well the Solver can assemble the required steps based on how many SQL elements appear in the original query. For our analysis, we used SQLGlot[3] to create an abstract syntax tree that shows the query's complexity. The number of nodes in an abstract syntax tree (AST) corresponds

---
[3] https://github.com/tobymao/sqlglot

to the number of components our Solver must interact with to arrive at an answer, which is proportional to the number of operations any Python program must also use. We designate queries by the number of AST nodes such that *Easy* is $\leq 8$, *Medium* is $\leq 15$, *Hard* is $\leq 20$, and *Extra Hard* is $> 20$. The experimental results for WikiSQL are provided in Table 2.

**Operator Difficulty** We also evaluate in Table 2 the accuracy of our approach by SQL aggregator, which includes SELECT, MAX, MIN, COUNT, SUM, and AVG operations. WikiSQL is relatively simple as reflected by $71.8\%$ of SELECT questions, with COUNT as the next prominent operator at $9.1\%$. The top operators are SELECT and SUM. In contrast, HiddenTables exposes gpt-3.5-turbo's deficiency in fetching extrema within a column with MIN/MAX or simple counting. AVG underperforms, as a significant number of tables include a grand total entry.

## 5.4 WikiTableQuestions

**Operator Difficulty** We tag each question in WikiTQ as a **Select**, **Filter**, **Aggregate**, **Superlative**, **Arithmetic**, **Comparative**, **Group** or **Other** operator, as inspired by (Liu et al., 2022), to further understand the limitations regarding gpt-3.5-turbo. Table 3 enumerates the operator types and the performance breakdown by split. In order to quickly tag each question, we used a 7-shot approach using one example per type of question, then leveraged gpt-3.5-turbo to generate the best category for the question. This provides insight into how the model handles each question during inference time, as the same assumptions in

categorizing the question influence the generated code.

## 5.5 SQA

**Dependency Difficulty** As a conversational dataset, SQA allows the profiling of `gpt-3.5-turbo`'s performance on follow-up questions. In Table 4, we denote the accuracy across several facets. We profile the overall accuracy for each sample and denote the accuracy for the sequence. For intermediate questions $Q_i$, we showcase the accuracy of the $i$-th question in the conversation. As expected, highly compositional questions tend to struggle more than initial sequence questions.

**Operator Difficulty** Since SQA builds off of compositional questions from WikiTQ, there is significant overlap between the two. Therefore, we reuse our generated 7-shot question taxonomies for all SQA samples found in the WikiTQ set. If not found, the category defaults to N/A (2,874 samples).

## 5.6 Privacy & Efficiency vs. Accuracy: Tradeoff

HiddenTables has demonstrated that in order to have full privacy and efficiency in the context of table question-answering, the lack of illustrative examples or the holistic table degrades accuracy. Privacy is a crucial concern when working with sensitive data, especially in industries that are highly regulated. By generating code derived only from the question and schema of a table, *rather than* the whole table, data exposure can be limited. Therefore, the Oracle, via the Secure Interpreter, only accesses the relevant portions of the data *on-premise*, mitigating the risk of any data leaks. HiddenTables compensates the substantial increase in difficulty from blindly solving TableQA by implementing the pair-programming iterative approach between the Solver and Oracle LLMs, as outlined in The Conversation (§3.3). This iterative approach to problem solving yields a $+6.7\%$ increase for WikiSQL, a $+8.2\%$ increase for WikiTQ, and a $+11.4\%$ increase in SQA.

Efficiency is another consideration regarding large knowledge bases or computationally intensive tasks. First, generating code allows systems to focus computational resources on subsets of the data internally, rather than processing the entire set as a multi-span extraction or aggregation prob-

lem. This results in fewer tokens required during the inference step of an LLM, resulting in lower latency and faster response times. Our approach used $48.5\%$ of the total tokens, if table contexts are considered. This proportion will decrease as table sizes increase in either rows or columns.

HiddenTables comes with a drawback in terms of accuracy. When relying solely on the schema, the problem shifts from a multi-span extraction task to a semantic parsing and code generation task. This added complexity requires LLMs to interpret and comprehend the question alongside the table structure. As a result, we see that HiddenTables's final accuracy is below TaPEx (Liu et al., 2022). By forcing LLMs to align the interpretation of queries to structure, errors in understanding the format of data dominates most failure cases. While additional conversation rounds mitigate this risk, other errors such as relying on extraction within a full text column still prove difficult.

## 6 Conclusion

In this work, we introduced a novel approach to evaluating the generalizability of LLMs across 3 table question-answering datasets. By creating a co-operative game that withholds the underlying data from the the model, HiddenTables challenges the Solver to make educated guesses via programmatic commands and operators to be in a state of successful retrieval. We have shown that this construct enables a computationally efficient large-scale testing of LLMs on massive datasets in tandem with ensuring the security of the tabular data. Also, our study provides insights that this task is considerably more difficult than traditional holistic models - yet lends itself to potentially large-scale industrial applications. We have also quantified this efficiency by showcasing the number of generated tokens in contrast with those of conventional models. We also contribute PyQTax, a dataset aligning generated python code to table questions and various taxonomies for 116,703 samples. Overall, our work provides a promising direction for future research in the field of table question-answering and has devised a novel construct in the deployment process of language models.

## Limitations

While our work presents a novel approach to evaluating the generalizability of LLMs on table-question answering datasets, it is imperative to dis-

cuss several limitations to our system. Foremost, our approach requires a Solver to generate code and answer the user query, which may be infeasible. Additionally, our system's reliance on programmatic commands and operators may result in a lack of flexibility when it comes to answering certain types of queries.

Next, while HiddenTables protects the information in the tables by withholding the underlying data from the LLM, it may not be able to address the issue of data privacy in cases that the table schema may contain sensitive information. Moreover, our system's reliance on an Oracle to evaluate the Solver's code may not be scalable in cases when there is a high volume in user queries.

Lastly, while our results demonstrate the effectiveness on English language datasets, its scalability to other languages with more complex morphologies and diacritics is an area that requires further investigation. Additionally, questions are tailored to each dataset, where WikiSQL questions reiterate column names to align language to table retrieval. The discrepancy between experimental questions and real-life user queries can be substantial and warrants further investigation. In summary, while our system presents a promising direction for future research in table question-answering, these limitations must be acknowledged to enable its wider adoption.

## Acknowledgements

This paper was prepared for informational purposes by the Artificial Intelligence Research group of JP-Morgan Chase & Co and its affiliates ("J.P. Morgan") and is not a product of the Research Department of J.P. Morgan. J.P. Morgan makes no representation and warranty whatsoever and disclaims all liability, for the completeness, accuracy or reliability of the information contained herein. This document is not intended as investment research or investment advice, or a recommendation, offer or solicitation for the purchase or sale of any security, financial instrument, financial product or service, or to be used in any way for evaluating the merits of participating in any transaction, and shall not constitute a solicitation under any jurisdiction or to any person, if such solicitation under such jurisdiction or to such person would be unlawful.

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

## A   Few-Shot Categorization of Questions

To provide better clarity into the generalizability of gpt-3.5-turbo, we breakdown WikiTQ into seven categories of questions. By using the same LLM and the Solver, we gain insight into how gpt-3.5-turbo recognizes and understands what kind of operations should be performed for a given question, based on semantics. To label each question, we select a representative example for each question category, and provide this as a 7-shot prompt to the model. We include the candidate question and a directive to label it, then parse and reconcile the generated category with the prescribed eight (Other is a fallback category). For SQA, there is an overlap between WikiTQ, and therefore we reuse the same labels when applicable. See Table 17 for an example of each question type, plus the semantic span that correlates with the category.

## B   Implementation: Secure Interpreter

To execute code generated by the Solver, we provided the Oracle a secure interpreter that can directly interact with the data *on-premise*. This means that our setup, in order to preserve privacy, is executed locally. The Solver's generated output is checked for any malicious code, in case of a potential attack through code injection or external requests. First, the interpreter is fire-walled to have no external connections, as the data is already *on-premise*. Second, the interpreter does not allow for any additional packages to be imported. The generated code is inspected for `import *`, `import * as *`, `from * import *` and replaced with an empty string. The namespace of the interpreter is pre-installed with verified packages. Finally, to avoid malicious code intended to erase or corrupt data, all operations are performed on a copy of the table. If a copy is not feasible, the database only allows for read operations. Any write or in-place operations on the source data are strictly denied. Intermediate artifacts are allowed to be manipulated during execution.

## C   Instructions for RISQ

We outline the instructions and the *(rationale)* in parenthesises.

1. You must write python code and operate on a pandas dataframe named `df`. *(Aligns Solver to the start variable to operate on)*

2. Use `reset_index()` after any *groupby* operation involving aggregation, and sort (Common error is to access an aggregated variable, yet Pandas stores these in the table's index)

3. All `.str.contains` MUST BE case insensitive (`case = False`). *(Helps improve the hit rate within a column when filtering)*

4. Do not use inplace operators - save each intermediate variable *(Improves chain of thought with code by saving each step as a new variable)*

5. The final answer must be saved as `final_answer` *(Easier to find the generated answer, although optional)*

6. Do not ask for clarification, you have everything you need to answer the question through the column headings *(Gives confidence to*

gpt-3.5-turbo *to directly answer the question and take risks)*

7. You cannot look at the data - just write code instead *(Reiterates the main generation objective)*

8. If you think you cannot answer a question, look for column such as note, comments, that may contain the answer and return the row item. *(Guides* gpt-3.5-turbo *to attempt retrieval code on free-text columns, which may contain the answer)*

## D  Sample Solver Code

```
1  # Q1: What was the highest vote
2  #      number for Bush?
3  max_votes_bush = df['Bush#'].max()
4
5
6  # Q2: What is the Position of the
7  #      person with a birthplace of
8  #      phoenix, new yorsk?
9  final_answer = df[
10     df['Birthplace'].str.contains(
11         'phoenix, new york',
12         case=False)
13     ]['Position'].iloc[0]
14
15
16 # Q3: Which Home team has a Tie no
17 #      of replay, a Date of
18 #      19 november 1985, and an
19 #      Away team of tranmere rovers?
20 final_answer = df[
21     (df['Tie no'].str.contains(
22         'replay', case=False)) &
23     (df['Date']
24         == '19 november 1985') &
25     (df['Away team']
26         == 'tranmere rovers')
27     ]['Home team'].iloc[0]
```
Listing 1: Generated Python Code from Natural Language Questions (WikiSQL)

```
1  # Q1: What is the difference,
2  #      in years, between the
3  #      founding of the heidelberg
4  #      football club and the
5  #      eltham football club?
6  heidelberg_founded = df[
7      df['Club'].str.contains(
8          'heidelberg', case=False)
9      ]['Founded'].iloc[0]
10
11 eltham_founded = df[
12     df['Club'].str.contains(
13         'eltham', case=False)
14     ]['Founded'].iloc[0]
15
16 difference = abs(
17     int(heidelberg_founded)
18     - int(eltham_founded))
19 )
```

```
20
21
22 # Q2: Who was the top winner in 2002
23 #      of the division 1 undergraduate?
24 #
25 # Group by Year and Division I
26 # Undergraduate, and find the max
27 # of the Division I Overall
28 grouped = df.groupby(['Year',
29     'Division I Undergraduate'])
30     ['Division I Overall'].max()
31
32 # Reset the index
33 grouped = grouped.reset_index()
34
35 # Filter the result for the year 2002
36 # and sort by Division I Overall
37 filtered = grouped[
38         grouped['Year'] == '2002'
39     ].sort_values(
40         by='Division I Overall',
41         ascending=False)
42
43 # The top winner is the first row
44 # of the filtered result
45 final_answer = filtered.iloc[0][
46     'Division I Undergraduate'
47     ]
```
Listing 2: Generated Python Code from Natural Language Questions (WikiTQ)

```
1  # Q1: S1| Who are all the players?
2  all_players = df['Player'].unique()
3
4
5  # Q2: S1| What event did an American
6  #          get a medal in?
7  medal_winners = df[
8      df['Gold'].str.contains(
9          'USA|United States',
10         case=False)
11     | df['Silver'].str.contains(
12         'USA|United States',
13         case=False)
14     | df['Bronze'].str.contains(
15         'USA|United States',
16         case=False)]
17 final_answer = (medal_winners
18     ['Event'].iloc[0])
19
20
21 # Q3: S1| Which baseball players have
22 #          scored more than 500 home
23 #          runs?
24 #      S2| Of those, which have scored
25 #          less than 600?
26 #      S3| Who among those players
27 #          reached the mark of 500 home
28 #          runs in the month of
29 #          september of any year?
30 #      S4| Of these who has the least
31 #          home runs?
32 more_than_500 = df[
33     df['HR'].astype(int) > 500]
34
35 less_than_600 = more_than_500[
36     more_than_500['HR'].astype(int)
37     < 600]
38
```

```
39  september_500 = df[
40      df['Date reached 500 HR']
41      .str.lower()
42      .str.contains('september',
43          case=False)]
44
45  least_home_runs = less_than_600.loc[
46      less_than_600['HR'].astype(int)
47      .idxmin(), 'Player']
```

Listing 3: Generated Python Code from Natural Language Questions (SQA)

## E  Common Error Codes

| Error Code | Count | Percent |
|------------|------:|--------:|
| No Code Provided | 4,573 | 51.17 |
| IndexError | 1,254 | 14.03 |
| AttributeError | 865 | 9.68 |
| ValueError | 715 | 8.00 |
| KeyError | 567 | 6.34 |
| IndentationError | 346 | 3.87 |
| NameError | 300 | 3.36 |
| SyntaxError | 160 | 1.79 |
| TypeError | 102 | 1.14 |
| DateParseError | 14 | 0.16 |
| OutOfBoundsDatetime | 12 | 0.13 |
| IndexingError | 12 | 0.13 |
| RedefinitionError | 6 | 0.07 |
| IntCastingNaNError | 4 | 0.04 |
| UndefinedVariableError | 3 | 0.03 |
| FileNotFoundError | 2 | 0.02 |
| ModuleNotFoundError | 2 | 0.02 |

Table 6: **Common error codes encountered during HiddenTables, with proportional percentages. Total number encountered: 8,937.**

We outline common errors in Table 6. The largest failure case was `gpt-3.5-turbo` not providing any executable code. This usually occurs when a question does not aligning with any column names. Furthermore, `IndexError` exclusively occurs at attempting to directly access a table value that is strictly out of bounds for the index, which is expected if the Solver does not know how many records are contained in the table after a **Filter**, **Comparative**, or **Superlative** operator. The next most common issue was an `AttributeError`, often triggered by `gpt-3.5-turbo` being unable to infer the correct type of variable the code operates on. For instance, the most common objection of the interpreter was "Can only use .str accessor with string values!" indicating a failure to correctly apply string methods onto a pandas dataframe. `ValueError` arose when boolean indexing that had NA / NaN values - of which a fix is to include .str.contains(*, `na=False`). Finally, `KeyError` is fairly straightforward - the

Solver produced code that accesses a column not available in the transformed tables, either through hallucination or as a byproduct of aggregation.

## F  Examining the Effect of Table Size on Performance

| Partition | Small | Average | Large |
|-----------|------:|--------:|------:|
| **Easy** | 86.2 | 78.3 | 73.6 |
| **Medium** | 73.8 | 72.7 | 69.7 |
| **Hard** | 58.8 | 59.6 | 59.8 |
| **Extra Hard** | 53.5 | 56.0 | 55.0 |
| SELECT | 74.0 | 72.6 | 69.2 |
| COUNT | 64.4 | 60.9 | 57.8 |
| MAX | 51.8 | 57.0 | 59.5 |
| MIN | 55.1 | 60.0 | 58.7 |
| AVG | 41.3 | 44.9 | 43.7 |
| SUM | 69.0 | 68.4 | 68.8 |

Table 7: Cross-taxonomy accuracy for all **WikiSQL** sets by difficulty and operator against table size (tokens). For difficulty, we see performance degrade as the overall table size increases.

| Partition | Small | Average | Large |
|-----------|------:|--------:|------:|
| **Aggregate** | 50.2 | 45.0 | 42.3 |
| **Filter** | 48.1 | 41.2 | 35.5 |
| **Superlative** | 43.3 | 36.0 | 36.7 |
| **Comparative** | 46.2 | 39.8 | 40.8 |
| **Select** | 40.2 | 38.9 | 36.2 |
| **Arithmetic** | 36.2 | 32.4 | 29.4 |
| **Other** | 25.0 | 53.8 | 46.2 |
| **Group** | 55.6 | 41.5 | 41.2 |

Table 8: Cross-taxonomy accuracy for all **WikiTQ** sets by operator against table size (tokens). Generally, performance decreases as tables grow larger in tokens.

| Partition | Small | Average | Large |
|-----------|------:|--------:|------:|
| **$Q_1$** | 52.9 | 53.6 | 51.3 |
| **$Q_2$** | 35.9 | 32.5 | 32.3 |
| **$Q_3$** | 40.4 | 33.8 | 31.8 |
| **$Q_4$** | 41.5 | 39.0 | 37.8 |
| **$Q_5$** | 43.9 | 33.9 | 45.8 |
| **$Q_6$** | 14.3 | 50.0 | 18.2 |
| **$Q_7$** | 50.0 | 33.3 | 40.0 |
| **$Q_8$** | 0.0 | 0.0 | 50.0 |
| **Aggregate** | 38.9 | 34.3 | 23.6 |
| **Filter** | 42.7 | 42.0 | 41.9 |
| **Superlative** | 46.2 | 39.8 | 44.2 |
| **Comparative** | 42.4 | 41.2 | 34.3 |
| **Select** | 42.3 | 37.2 | 39.7 |
| **Arithmetic** | 22.7 | 35.2 | 40.2 |
| **Other** | 55.6 | 46.7 | 58.8 |
| **Group** | 45.5 | 55.6 | 39.4 |
| **N/A** | 43.2 | 42.5 | 42.5 |

Table 9: Cross-taxonomy accuracy for all **SQA** sets by question sequence and operator against table size (tokens). Generally, performance decreases as table size increases, with a few exceptions.

# G  Examining the Effect of the Number of Rows on Performance

| Partition | Small | Average | Large |
|---|---|---|---|
| **Easy** | 80.1 | 79.1 | 75.3 |
| **Medium** | 71.9 | 72.7 | 72.5 |
| **Hard** | 58.7 | 58.6 | 61.5 |
| **Extra Hard** | 55.4 | 53.8 | 56.2 |
| SELECT | 72.8 | 72.1 | 69.2 |
| COUNT | 61.4 | 61.2 | 60.0 |
| MAX | 55.0 | 55.0 | 59.4 |
| MIN | 58.9 | 57.7 | 58.5 |
| AVG | 42.7 | 43.4 | 45.2 |
| SUM | 69.7 | 67.0 | 69.6 |

Table 10:  Cross-taxonomy accuracy for all **WikiSQL** sets by difficulty and operator against the number of table rows. There are no discernible trends, highlighting that HiddenTables is not dependent on the number of rows for performance. Therefore, the trends in Table 7 are exclusively driven by the number of columns.

| Partition | Small | Average | Large |
|---|---|---|---|
| **Aggregate** | 44.3 | 45.9 | 43.9 |
| **Filter** | 43.1 | 40.3 | 37.7 |
| **Superlative** | 40.0 | 35.5 | 38.5 |
| **Comparative** | 40.7 | 43.2 | 39.5 |
| **Select** | 36.1 | 39.3 | 38.7 |
| **Arithmetic** | 33.5 | 32.7 | 29.5 |
| **Other** | 36.4 | 60.0 | 44.4 |
| **Group** | 37.5 | 47.6 | 33.3 |

Table 11:  Cross-taxonomy accuracy for all **WikiTQ** sets by operator against the number of table rows. Generally, performance for most partitions is greatest on average sized tables.

| Partition | Small | Average | Large |
|---|---|---|---|
| **$Q_1$** | 53.4 | 51.5 | 54.5 |
| **$Q_2$** | 34.1 | 32.3 | 34.4 |
| **$Q_3$** | 36.5 | 34.1 | 34.7 |
| **$Q_4$** | 42.1 | 35.7 | 41.8 |
| **$Q_5$** | 37.5 | 34.0 | 48.1 |
| **$Q_6$** | 28.6 | 40.0 | 27.3 |
| **$Q_7$** | 40.0 | 40.0 | 40.0 |
| **$Q_8$** | 0.0 | 0.0 | 50.0 |
| **Aggregate** | 34.2 | 33.5 | 31.7 |
| **Filter** | 42.0 | 40.7 | 45.1 |
| **Superlative** | 42.7 | 40.1 | 46.2 |
| **Comparative** | 42.5 | 39.1 | 37.7 |
| **Select** | 43.4 | 36.5 | 35.6 |
| **Arithmetic** | 28.9 | 32.3 | 48.5 |
| **Other** | 38.5 | 60.7 | 36.4 |
| **Group** | 45.5 | 55.6 | - |
| **N/A** | 41.6 | 42.0 | 45.7 |

Table 12:  Cross-taxonomy accuracy for all **SQA** sets by question sequence and operator against the number of rows. **Filter** increases in performance, perhaps being agnostic to the number of items, while **Aggregate** shows increased sensitivity to the inclusion of outliers.

# H  Examining the Effect of the Number of Columns on Performance

| Partition | Small | Average | Large |
|---|---|---|---|
| **Easy** | 78.8 | 77.7 | 80.6 |
| **Medium** | 75.9 | 72.7 | 64.0 |
| **Hard** | 61.2 | 59.0 | 54.8 |
| **Extra Hard** | 58.3 | 54.8 | 49.9 |
| SELECT | 74.5 | 72.1 | 69.2 |
| COUNT | 60.4 | 62.8 | 58.1 |
| MAX | 57.3 | 55.9 | 54.7 |
| MIN | 57.2 | 57.9 | 61.0 |
| AVG | 51.6 | 40.6 | 29.7 |
| SUM | 68.3 | 69.8 | 66.3 |

Table 13:  Cross-taxonomy accuracy for all **WikiSQL** sets by difficulty and operator against the number of table columns. As difficulty increases, the number of table columns has more influence on performance, yet for simple questions shows no differentiation. No discernible trend can be inferred for SQL operator.

| Partition | Small | Average | Large |
|---|---|---|---|
| **Aggregate** | 45.8 | 45.4 | 42.6 |
| **Filter** | 40.1 | 41.2 | 40.0 |
| **Superlative** | 37.6 | 40.5 | 32.2 |
| **Comparative** | 38.8 | 41.6 | 46.2 |
| **Select** | 38.8 | 38.0 | 36.6 |
| **Arithmetic** | 29.5 | 33.1 | 35.0 |
| **Other** | 49.1 | 53.1 | 33.3 |
| **Group** | 38.7 | 41.7 | 41.2 |

Table 14:  Cross-taxonomy accuracy for all **WikiTQ** sets by operator against the number of columns. Performance increases with more columns, suggesting that question complexity plays a greater role than operator.

| Partition | Small | Average | Large |
|---|---|---|---|
| **$Q_1$** | 53.5 | 51.8 | 53.4 |
| **$Q_2$** | 33.1 | 33.6 | 33.2 |
| **$Q_3$** | 35.4 | 34.4 | 35.4 |
| **$Q_4$** | 42.1 | 36.9 | 38.7 |
| **$Q_5$** | 40.5 | 38.1 | 44.0 |
| **$Q_6$** | 30.8 | 31.6 | 50.0 |
| **$Q_7$** | 25.0 | 37.5 | 66.7 |
| **$Q_8$** | 0.0 | 25.0 | 0.0 |
| **Aggregate** | 37.5 | 33.6 | 27.0 |
| **Filter** | 41.6 | 41.3 | 45.8 |
| **Superlative** | 40.6 | 43.4 | 42.5 |
| **Comparative** | 41.8 | 38.5 | 37.8 |
| **Select** | 37.4 | 39.1 | 37.5 |
| **Arithmetic** | 38.3 | 31.0 | 39.2 |
| **Other** | 38.3 | 47.8 | 58.8 |
| **Group** | - | 45.5 | 55.6 |
| **N/A** | 44.9 | 39.6 | 44.2 |

Table 15:  Cross-taxonomy accuracy for all **SQA** sets by question sequence and operator against the number of columns. There is no discernible influence of columns on the performance of HiddenTables.

| | OP | Count | Query | Answer |
|---|---|---|---|---|
| **Easy** | COUNT | 1 | Name the number of Februaries | 12 |
| | MAX | 264 | What was the highest home total? | 91143 |
| | MIN | 345 | List the lowest number of assists | 10 |
| | SELECT | 8 | What are names of the episodes that airs at 2:00pm? | [···] |
| **Medium** | AVG | 1,208 | Name the average attendance with Dallas visitor | 16859 |
| | COUNT | 5,132 | For how many years was the urban population 57%? | 4 |
| | MAX | 2,386 | What was the largest crowd where the home team was Fitzroy? | 18000 |
| | MIN | 2,291 | What year was the tournament first held in Italy? | 1930 |
| | SELECT | 43,512 | What are all the results for New York 7th district? | re-elected |
| | SUM | 1,130 | How many specimens does the SANT instituion have? | 71000 |
| **Hard** | AVG | 1,407 | Name the average pop for Chūgoku and prefecture of Okayama | 700646 |
| | COUNT | 1,658 | What is the total when silver is < 1 and rank > 3? | 1.0 |
| | MAX | 1,514 | What is Honda's highest grid with a time of +1:38.407? | 7 |
| | MIN | 1,489 | What was the lowest total for Italy when the latest win is 1950? | 7 |
| | SELECT | 11,893 | What was the tonnage of the Great Britain ship Batna? | 4399 |
| | SUM | 1,342 | What is the number of points ofr East Germany, and Places of 88? | 170.54 |
| **Extra Hard** | AVG | 520 | Tell me the average rank for losses < 6 and wins < 11 for michigan state. | 10 |
| | COUNT | 556 | How many years did the NY Giants win with a result of 15-7 at Lincoln Financial? | 1 |
| | MAX | 475 | What is the latest year Rafael Nadal was in the French Open, Roger Federer in Wimbledon, and Roger Federer in the Australian Open? | 2007 |
| | MIN | 506 | What is the lowest total with rank <8, a silver > 6, and 20 as the bronze? | 57 |
| | SELECT | 2510 | Which Production in 2009 had a Result of Nominated at the Helpmann awards Award Ceremony? | wicked |
| | SUM | 477 | What is the sum of Population when Area < 5131, Pop Density < 180, Subdivisions is Parishes, and Capital is Roseau? | 72660 |

Table 16: Cross-taxonomy samples for all **WikiSQL** sets by difficulty and operator. The diversity in answer types warrants a flexible approach to table QA through code. We hope enumerating samples without the corresponding tables proves that HiddenTables is a difficult game, and the Solver may have to make several attempts before a successful retrieval is made by the Oracle.

| OP | Query | Answer |
|---|---|---|
| **Aggregate** | Before 1999, how **many** series occurred? | 6 |
| | How **many** species of birds are there in Guatemala? | 684 |
| | What is the **total amount** of students who took the test in 2007? | 97136 |
| **Filter** | Who was the only Candidate with the hometown of **Tulsky**? | Alissa Joanndova |
| | Name all the nations that won at least **five silver** medals. | Puerto Rico |
| | Which song charted **in the US** but **not the UK**? | Set the Night to Music |
| **Superlative** | What opponent is at the **top** of the Chart? | Japan |
| | Which country took the **least** amount of time? | United States |
| | Which team was the **runner up** the most times? | Arsenal |
| **Comparative** | Which event occurred first: **St. Paul Open or the Charlotte Open**? | Charlotte Open |
| | Which nation won the same number of gold medals **as Hungary**? | Bulgaria |
| | What country had the least amount of drivers, **Germany or the UK**? | Germany |
| **Select** | What is the **name** of the first venue on this list? | Riverside Montien Hotel |
| | In what **country** is Bologna? | Italy |
| | Which **1965 film** starred actors Elizabeth Taylor & Richard Burton? | The Sandpiper |
| **Arithmetic** | How **many more** AM channels are there than FM channels? | 9 |
| | What is the **difference** of weight between the Maria Bell & the Carolus Bell? | 3145 |
| | What was the **difference**, in time, between the first place competitor and the third place competitor? | +0.400 |
| **Group** | **For each** winning game, what was their score? | 6-1 |
| | What was the ranking **in each** November game? | #2 |
| | Name **all winners** of the Caribbean Cup | Trinidad & Tobago |
| **Other** | What is **next after** chuchillo -2? | Solano - 3 |
| | What was the **first** outcome listed on this chart? | Winner |
| | What is the **first** name ranked? | Alberto García |

Table 17: Taxonomy samples for **WikiTQ** generated through our 7-shot classification procedure, outlined in the Appendix §A. We also **highlight key semantics** within each sample that aligns for the category.