# OpenReview forum: "HiddenTables and PyQTax: A Cooperative Game and Dataset For TableQA to Ensure Scale and Data Privacy Across a Myriad of Taxonomies"
_EMNLP/2023/Conference — EMNLP 2023 Main_

### Official Review · Reviewer_6csL · 2023-08-03

**Soundness:** 3

**Excitement:**

3: Ambivalent: It has merits (e.g., it reports state-of-the-art results, the idea is nice), but there are key weaknesses (e.g., it describes incremental work), and it can significantly benefit from another round of revision. However, I won't object to accepting it if my co-reviewers champion it.

**Missing References:**

Binding Language Models in Symbolic Languages: https://arxiv.org/abs/2210.02875

**Paper Topic And Main Contributions:**

This paper proposes a novel adversarial framework called "HiddenTables" for evaluating the ability of large language models (LLMs) to solve table question answering (QA) tasks without exposing the full table content to the model. The key ideas are:

   - An "Oracle" agent that holds the table data and interacts with the LLM "Solver" agent via a restricted interface by providing schema and natural language instructions.
   - The Solver must generate executable Python code to retrieve answers, based solely on the schema and instructions from the Oracle. This tests the LLM's ability to map language to solutions without seeing real table data.
   - A secure "conversation" protocol between Oracle and Solver to allow iterative refinement of generated code.

The main contributions are:
- The HiddenTables framework itself, which provides privacy and efficiency benefits compared to exposing full tables to LLMs.
- Analysis of limitations of LLMs on table QA using HiddenTables, showing degraded accuracy but more interpretability.
- A new dataset PyQTax with 116k samples linking questions to Python code and taxonomies.


**Questions For The Authors:**

1. Can you provide a more detailed breakdown of accuracy by table size and question complexity? Where does the Solver struggle most?
2. How do the accuracy results compare if the full table is exposed to the LLM? Is there a way to quantify the privacy/efficiency vs accuracy tradeoff?
3. How scalable is the iterative Oracle/Solver conversation protocol? What are the limits on number of queries before performance degrades?

**Reasons To Accept:**

- The idea of restricting table access for LLMs to test true language understanding is insightful. The proposed framework is well-motivated.
- Thorough experiments across diverse datasets (WikiSQL, WikiTableQuestions, SQA) validate the utility of HiddenTables.
- The framework enables large-scale testing of LLMs on massive datasets not possible with standard approaches, with built-in privacy

**Reasons To Reject:**


-  The accuracy degradation under HiddenTables, while expected, seems quite significant on complex questions. More analysis on where the Solver struggles would strengthen the findings.
- It's not clear if the gains in privacy and efficiency outweigh the loss in accuracy for practical applications. Some discussion of this tradeoff would be useful.
- The scalability of the proposed conversation protocol between Oracle and Solver needs more analysis, especially for high query volumes.



**Reproducibility:**

3: Could reproduce the results with some difficulty. The settings of parameters are underspecified or subjectively determined; the training/evaluation data are not widely available.

**Reviewer Confidence:**

4: Quite sure. I tried to check the important points carefully. It's unlikely, though conceivable, that I missed something that should affect my ratings.

---

> ### Author Rebuttal · Authors · 2023-08-28
>
> Thank you for your thorough review of our paper and appreciate your valuable feedback. In addition, thank you for bringing the missing reference to our attention and will ensure that the reference to "Binding Language Models in Symbolic Languages” is included in the revised manuscript. Below, we have considered your comments:
> 1. Accuracy Degradation: In our revised version, we will include a more detailed breakdown of accuracy by table size versus question complexity. In addition, we are happy to expand the discussion section regarding on the accuracy degradation for complex questions that relied on multi-hop reasoning (WikiSQL, WikiTables) or prior solutions (SQA) - which can suffer for error propagation if prior steps are wrong. For example, in WikiSQL Easy, the accuracy for Small is 96%, Average is 81%, and Large is 74%. We can provide this in any number of ways to highlight degradation or stagnant performance.
> 2. Privacy and Efficiency Tradeoff: We are happy to add a comprehensive discussion of this tradeoff between accuracy and privacy, and relating it to real world examples. We will also aim to provide a clearer understanding of the practical implications of utilizing the HiddenTables framework in regards to PPI, HIPA, criminal information, trade secrets, or FERPA.
> 3. Scalability of “Oracle”/“Solver” Conversation Protocol: We are happy to expand on this by analyzing the token usage, latency as conversation length grows, and generation of new tokens.
> 4. Question 1) Accuracy Breakdown by Table & Complexity: We can add a cross section between any dimension and permutation in the appendix to highlight the struggles by the solver. Each solution is tied to multiple taxonomies - allowing for any comparison. We will include this in the appendix in the revised version.
> 5. Question 2) Full Information Accuracy: We will conduct experiments in our revised version where the full table is exposed to the LLM.
> 6. Question 3) Scalability:  We limit the conversation to 7 chat turns, and for each dataset we enumerate the total number of solutions that are correct per turn. We will add a section to address scalability, in addition to breaking down table size by rows/columns and the number of conversation cycles. We have already analyzed token consumption and can expand on this section with latency and timing information.

---

### Official Review · Reviewer_xnAr · 2023-08-04

**Soundness:** 3

**Excitement:**

4: Strong: This paper deepens the understanding of some phenomenon or lowers the barriers to an existing research direction.

**Paper Topic And Main Contributions:**

The paper tackles the complex challenges faced by Large Language Models (LLMs) in contextually analyzing table question-answering tasks, proposing "HiddenTables" - an alternative method, which involves generating code to solve Table QA tasks using only the schema of the table. This approach is computationally faster and more cost-effective while simultaneously mitigating the common problems associated with tokenization (table parsing etc.). Furthermore, it eliminates the risk of private data leakage and includes inherent mechanisms for interpretability.




**Questions For The Authors:**

1) "Our adversarial game is a robust demonstration that the accuracy of gpt-35-turbo decreases rapidly throughout when language". How exactly has this claim been shown?

2) Do the variations within the "game" prompt rapidly change the results of the experimentation? Have the authors tried testing the method for sensitivity, as it would highly impact the interpretable component?

3) Can the method be tested against baselines, i.e. CoT in zero/few-shot settings? Such an analysis can establish the advantages of the approach more clearly.

**Reasons To Accept:**

This paper contributes valuable insights and novel solutions to the challenges in TableQA tasks. The content is solid, and with some refinement in language, it would be a strong addition to the conference. The method allows for the simultaneous solution of computational, data leakage and tokenization issues prevalent throughout the TableQA tasks. The offered mechanism allows for a more interpretable family of solutions while maintaining the cost-effective nature of the method. The provided ablations and qualitative analysis is helpful for understanding the complexity of the task. A dataset of code-aligned queries/table/code/answer touples is also provided.



**Reasons To Reject:**

1) [Minor] The choice of some terms and notations may need reconsideration as it was hard to follow. To more accurately reflect the approach and align better with the reader's understanding of the subject matter, wording such as "oracle" and "adversarial game" can be reworded. In particular, as the iterative generation approach does not involve an actual adversary or adversarial generation, it is confusing to the reader to understand the scope/terminology within the paper.

2) lines [95-97] "Our adversarial game is a robust demonstration that the accuracy of gpt-35-turbo decreases rapidly throughout when language". This point has not been widely explored throughout the paper and requires elaboration.

**Reproducibility:**

4: Could mostly reproduce the results, but there may be some variation because of sample variance or minor variations in their interpretation of the protocol or method.

**Reviewer Confidence:**

4: Quite sure. I tried to check the important points carefully. It's unlikely, though conceivable, that I missed something that should affect my ratings.

---

> ### Author Rebuttal · Authors · 2023-08-28
>
> We thank the reviewer for their feedback and their engagement with our work. Moreover, the reviewer’s expertise has deepened our research and we greatly appreciate it.  We have further clarified the specified comments below:
> 1. Choice of terms and notations: We will carefully review the terminology used throughout the paper, including terms such as "oracle" and "adversarial game," and make necessary revisions to ensure clarity and coherence as an interactive game.
> 2. Elaboration on the accuracy degradation: We are happy to clarify this contribution, as it relates to the decrease in accuracy compared to BART/BERT based models (TAPEX, TAPAS). As addressed in other reviews, we will also include an additional baseline to highlight this claim. The interactive game in our approach involves iteratively generating code to solve Table QA tasks, and the claim about the accuracy decreasing refers to the observation that without full information or example data, language models still struggle to align questions to commands. However, we do compare our approach against encoder-decoder methods with full information (TAPAS/TAPEX models).
> 3. Question 1) As stated above, our intention was to showcase that without “full information” such as TAPEX/TAPAS models, ChatGPT still struggles to solve TableQA tasks when provided only the schema of the data rather than the whole data. We will rephrase this contribution to highlight which baselines we are comparing our methodology with.
> 4. Question 2) Variations within the "game" prompt and their impact on the experimentation results: We have taken into consideration the sensitivity of our method. We have tested varying prompts on a sample sample set to evaluate the effects of prompt variations on the interpretable component. We will include a detailed discussion of these findings in the revised version of the paper, expanding on the Appendix Section “Instructions for RISQ.” Each prompt rule already has the motivation in parenthesis that explains our intuition from experimentation when constructing the final prompt.
> 5. Question 3) Alternative Baselines: We will include a discussion of further approaches to test against in the revised paper, alongside reporting our baselines for “full information” ChatGPT and single chat-turn conversations over the full interactive game. For the single ChatGPT attempt (schema only), the accuracy for WikiSQL is (61.4 Train, 62.2 Val, 61.35 Test) which is 6 below the interactive version. Additionally, WikiTQ is 31.7 train / 32.6 val / 33.6 test (8 points below the interactive) and SQA is 28.9 train / 27.7 val / 33.5 test (10 points below the interactive game).

---

### Official Review · Reviewer_fyQe · 2023-08-12

**Soundness:** 3

**Excitement:**

4: Strong: This paper deepens the understanding of some phenomenon or lowers the barriers to an existing research direction.

**Paper Topic And Main Contributions:**

This paper presents an adversarial approach “HiddenTables” for Table Question Answering which consists of two components, “Solver” and “Oracle”. The first one, “Solver”, predicts executable operations only relying on the table schema. This enables the usage of external LLM while protecting the table content. The “Oracle” executes the code from the “Solver” in a safe environment and passes the feedback. The proposed approach can potentially work with tables of arbitrary sizes. In addition, the authors propose a new large-scale dataset for Table Question Answering, PyQTax, which is built on top of the existing datasets but utilizes Python instead of SQL.

**Questions For The Authors:**

Question A: How the system performs in the case of the table contents available to the "Solver"? Is it still underperform TAPEX?

Question B: How does the Secure Interpreter check that the code received from the “Solver” is not malicious?

Question C: Why are you using the complexity of the original SQL query as a measure of the difficulty in PyQTax which is in Python? Do you have any evidence that the complexity of the solution in SQL matches that of Python?

**Reasons To Accept:**

- The authors propose an approach for TableQA which leverages the external LLMs while protecting the data in the tables, which would be applicable in real applications.
- The proposed method has no restrictions on the size of the table since only the schema of the table is passed to the model to generate the solution.
- Another contribution is a new dataset for TableQA with Python code instead of SQL that could be useful for future research.
- The paper is well-written and easy to follow, and the proposed approach is simple and easy to replicate.

**Reasons To Reject:**

- The main weakness of this paper is the lack of baselines. The authors experiment only with the proposed system based on the ChatGPT and BART-based TAPEX models, which does not show how certain components of the approach affect the quality. In particular, there is no baseline in which “Solver”, ChatGPT, has the full information about the table. Such a comparison would show how much more difficult the task is if the table content is not available for the model. Another missed baseline is ChatGPT as is (without any interactions) which would provide the basic performance on the TableQA task and probably show the benefits of the interactive systems like the proposed “Solver”-“Oracle”.

    Update: I acknowledge that the authors have provided a non-interactive ChatGPT baseline in their rebuttal. I encourage them to include these results in the next revision and add the "full information" baseline to provide a deeper understanding of the proposed system.
- The authors do not provide any details about the Secure Interpreter used in “Oracle” which can hinder reproducibility.

**Reproducibility:**

4: Could mostly reproduce the results, but there may be some variation because of sample variance or minor variations in their interpretation of the protocol or method.

**Reviewer Confidence:**

3: Pretty sure, but there's a chance I missed something. Although I have a good feel for this area in general, I did not carefully check the paper's details, e.g., the math, experimental design, or novelty.

---

> ### Author Rebuttal · Authors · 2023-08-28
>
> We thank the reviewer for engaging with our work and highlighting the key contributions our work presents to the community.  The reviewer’s feedback has provided great insights and we have great appreciation for this opportunity to further clarify any points.  In response to the raised points, we provide the following comments:
> 1. Lack of baselines: We focused on evaluating the proposed system based on the “Solver”/“Oracle" ChatGPT against BART-based TAPEX models to demonstrate its efficacy. However, we agree that including additional baselines would provide valuable insights for an end-to-end (full information) QA with ChatGPT. We will also expand the discussion surrounding the conversation rounds against a single round (schema-only) to demonstrate the benefits of the interactive system (by expanding section 5.2 Conversation Length & Accuracy). We will incorporate this baselines in our revised work to provide a comprehensive evaluation of our approach. For the single ChatGPT attempt (schema only), the accuracy for WikiSQL is (61.4 Train, 62.2 Val, 61.35 Test) which is 6 below the interactive version. Additionally, WikiTQ is 31.7 train / 32.6 val / 33.6 test (8 points below the interactive) and SQA is 28.9 train / 27.7 val / 33.5 test (10 points below the interactive game). Hence compared to a single round, our interactive game demonstrates the efficacy of improved performance.
> 2. Lack of details about the Secure Interpreter: We agree that additional provision of details regarding the Secure Interpreter used in the "Oracle" component would be beneficial for our research. Our details for the secure interpreter will be expanded from lines 201-207, and for additional clarification, “the secure interpreter is a firewall python interpreter that does not have any external internet connections.”
> 3. Question A) Performance of the system when the table contents are available to the “Solver": We will include an analysis of this scenario in our revised work, to highlight the challenges posed by limited table information.
> 4. Question B) Malicious Code & the Secure Interpreter: In the revised version, we will expand lines 201-207 with “The Secure Interpreter used in our system is a subprocess python shell, isolated within a virtual environment, and strictly prohibits write or external request access, ensuring its integrity and preventing any potential damage.” We will also expand implementation details in Python in the appendix.
> 5. Question C) Complexity of the original SQL as difficulty for Python: Considering the similarities of operations between Python/Pandas and SQL, we believe the complexity measure based on the abstract syntax tree of the ground truth SQL query serves as a reasonable proxy for assessing the difficulty of Python solutions in PyQTax.

---

### Meta-Review · Area_Chair_soiY · 2023-09-17

**Recommendation:** 4

**Metareview:**

The proposed methodology is novel, interesting and also timely (because it involves application of instruction-tuned LLMs for table-QA). The reviewers have pointed out to the absence of several meaningful baselines/ablations, which the authors have acknowledged and promised to include in the revised version of the paper. Overall, the suggestions of the reviewers, if incorporated, will make the paper a good contribution in EMNLP. I would strongly encourage the authors to add the suggested baselines and the analysis (e.g., the analysis on detailed breakdown of accuracy by table size and question complexity).

---

### Decision · Program_Chairs · 2023-10-07

**Decision:**

Accept-Main

**Comment:**

The proposed methodology is novel, interesting and also timely (because it involves application of instruction-tuned LLMs for table-QA). The reviewers have pointed out to the absence of several meaningful baselines/ablations, which the authors have acknowledged and promised to include in the revised version of the paper. Overall, the suggestions of the reviewers, if incorporated, will make the paper a good contribution in EMNLP. I would strongly encourage the authors to add the suggested baselines and the analysis (e.g., the analysis on detailed breakdown of accuracy by table size and question complexity).